# Homelessness at the San Diego Central Library: Assessing the Potential Role of Social Workers

**DOI:** 10.3390/ijerph19148449

**Published:** 2022-07-11

**Authors:** Lianne A. Urada, Melanie J. Nicholls, Stephen R. Faille

**Affiliations:** 1School of Social Work, San Diego State University (SDSU), San Diego, CA 92182, USA; mnicholls7003@sdsu.edu (M.J.N.); stephenalt@icloud.com (S.R.F.); 2Department of Medicine, University of California San Diego (UCSD), La Jolla, CA 92093, USA

**Keywords:** libraries, homelessness, substance use, opioid use, mental health, treatment, social work, public health crisis, human trafficking

## Abstract

Nationwide, public libraries are experiencing an increase in “on-premise” opioid overdoses and other issues (e.g., suicide attempts) affecting unstably housed library users. The public library presents a unique opportunity to access an otherwise hidden population. In partnership with the San Diego Central Library, researchers led focus groups, in-depth interviews, and surveys with 63 library patrons experiencing homelessness or housing instability (*n* = 49) and library staff (*n* = 14) (January–June 2019). Using a consensus organizing framework and mixed methods approach, the researchers conducted in-depth interviews exploring the library’s strengths and opportunities for patrons experiencing homelessness, the barriers to meeting their aspirations, and whether having a social worker at the library or other policy changes in government or the library could help. Specifically, participants answered inquiries about the opportunities for the library to address substance use and human trafficking. In brief surveys, library patrons and staff provided views on the patrons’ educational needs, library staff’s training needs, and changes needed in government or library policies. Results revealed the desire of the library patrons (69%) and staff (93%) to have a library social worker who could link patrons to housing services, substance use harm reduction or treatment, and address food-insecure youth/families and human trafficking/sexual exploitation. Participants also valued peer advocates with lived homelessness experiences. Over 70% of the unstably housed patrons said they would like library patrons to participate in peer leadership training. Other significant themes were the need for crisis prevention and intervention, connecting patrons to resources and each other, and creating consistent assistance. Libraries urgently need more on-premise support to address patrons’ pressing housing, health, and mental health needs.

## 1. Introduction

Nationwide, libraries are increasingly recognized as public spaces with the potential to assist patrons experiencing homelessness and housing instability [1,2,3,4,5,6,7,8,9,10,11]. At one northern Californian library, 15% of the patrons (of 5000 who entered daily) were homeless [12]. Over 160,000 people were homeless in California, representing nearly a quarter of those experiencing housing insecurity in the nation [13]. In addition, San Diego, southern California, ranked 4th in the United States for having the largest population of unhoused persons [14]. According to the San Diego Regional Task Force of the Homeless, 8102 were unstably housed in January 2020; in January 2022, it nearly doubled in downtown San Diego from a year earlier [15]. The library presents an opportunity to meet these patrons “where they are.”

However, library staff may not always be prepared to handle the myriad of social challenges their patrons face, such as opioid overdoses and mental health crises occurring on-premise [16]. In addition, people with housing instability lack adequate resources for the multiple risks they often encounter—food insecurity, substance use, and mental disorders [17,18,19,20].

A growing body of literature examines libraries’ changing role in meeting their vulnerable patrons’ needs [1,10]. Researchers conducted a scoping review on the role of public libraries in advancing population health [1]. For example, Toronto and San Francisco libraries provided food bank services during the COVID-19 pandemic [7]. Library directors in Pennsylvania said they needed additional training to adequately prepare for their patrons’ health and social issues; 12% witnessed an opioid overdose in the past year [3]. Another research team surveyed library staff and patrons throughout an extensive library system in Indiana [9,11,21]. In addition, the study identified ways a social worker could help libraries: being a broker by connecting libraries with outside agencies, conducting training for patrons and staff, and creating social work internships in libraries [21].

At least thirty other libraries in the U.S. have added social workers and other social service professionals to their teams [22,23,24]. For example, the San Francisco Public Library, the first in the nation to hire a social worker, is a model that views homelessness from a holistic angle. After hiring a clinical social worker, more than 150 people found permanent housing over six years, and 800 received other services [12,25]. The San Francisco Library also incorporated a Peer Support Model that included two Health and Safety Associates and former homeless patrons from the library. In Chicago, Amita Health System funded social workers at local libraries through donations from employees and money set aside for community programs to justify their tax-exempt status; they saw the need for behavioral and mental health treatment [25]. In San Diego, services co-located at libraries [1] included past onsite social workers, veterans’ services, health and wellness activities, and employment assistance.

However, few studies until recently partnered with libraries to interview patrons in-depth about the potential solutions to the escalating opioid and mental health crises at libraries. Few have documented this phenomenon as the opioid crisis converges at libraries [16,26]. Nor are there enough policies or evidence-based interventions to adequately address these issues in these public spaces. Using a consensus organizing framework [26] focusing on strengths and assets, this mixed methods research study conducted in-person, in-depth interviews, focus groups, and surveys with library patrons and staff housed at the library. The purpose was to gain insights into the lived experiences, barriers of those struggling with homelessness, and the opportunities and strengths the library setting presents. The goal was also to explore the prevalence of opioid overdoses and human trafficking around the library and the potential role of a library social worker.

## 2. Materials and Methods

### 2.1. The Setting

The San Diego Public Library’s (SDPL) Central Library, with nine floors, attracts patrons from every walk of life. The library’s vision is to be a “place for opportunity, discovery, and inspiration.” According to the library, 3000–4000 people enter daily through the library entrance. In addition to families, students, and residents, the library provides services to any visitor, regardless of their legal or criminal background or history of mental health or substance use. In addition, they shared that dozens of patrons with housing instability eagerly “line up waiting for the library to open each morning [18].” Library staff and patrons estimated that as many as 80% entering the library are those experiencing unstable housing.

Librarians at the SDPL Central Library expressed an urgency to better address the needs of their patrons experiencing homelessness, mental health, and substance use crises. Because unstably housed patrons could not bring all their belongings into the library, the patrons took advantage of the existing layout of the library space by sitting at computers where they could watch from large windows overlooking the street where their belongings were left. According to staff, a patron alerted them about a patron threatening to jump off the library’s highest floor to end his life. Other patrons overdosed in the library restrooms several times a month, lacking Narcan (treatment for opioid overdoses). To address these concerns, the San Diego State University School of Social Work faculty developed a research partnership and social work internship at the library to brainstorm solutions.

### 2.2. Participants

All unstably housed library patrons within the downtown SDPL Central Library or outside its perimeter were eligible to participate in the research. For the focus groups, they had to know someone experiencing homelessness if they were not homeless themselves. A total of 63 unduplicated individuals (49 library patrons and 14 service providers/library staff) completed the brief surveys. In-depth interviews were conducted with 36 patrons, all actively experiencing homelessness, and ten library staff. In addition, four focus groups (*n* = 29 patrons) were held; 19 patrons were unduplicated ones who did not choose to be interviewed individually. Details of the recruitment of the patrons and the staff are included below.

The social work faculty researcher, doctoral students in the Interdisciplinary Research on Substance Use, and master’s students in social work and public administration interviewed the participants. In addition, they had training on data collection and research practices (e.g., evidence-based research, community-based participatory research, consensus organizing, and community mobilization). See the interview guide and brief survey (Appendix A and Appendix B). The San Diego State University’s Institutional Review Board approved the protocol.

### 2.3. Recruitment of Library Patron Participants

Focus group participants responded to posters/flyers in the library. Library patrons could sign up at the third-floor library reference desk if they expressed interest in the focus group or an individual interview. The focus groups were held near the main entrance of the library’s first floor in a conference room (reserved ahead of time by a library manager).

The first focus group was held in February 2019 with 12 individuals who responded to fliers posted inside the library that solicited anyone who knew or had experienced the issue of housing instability. The research team also recruited the patrons at the library entrance as the library opened the same morning. Others responded via intercom announcements by the library manager before the group started.

The research team recruited a second group of seven individuals from the streets outside the library. Doctoral students with lived experiences or who worked with the population invited the participants if they were actively using opioids. They were skilled at detecting who might be using opioids and approaching small groups on the street. They provided a copy of the flier and encouraged them to come to the conference room that morning for the focus group. Identities were kept confidential, even from the library staff.

The third group (*n* = 5) held in April 2019 specifically recruited women or women living with children who faced housing instability. Again, fliers specifically recruiting women were posted. Some women in the focus group participated in the individual interview afterward and actively referred other women from their shelter.

During the focus group, participants signed up voluntarily for individual interviews to be scheduled and administered one-on-one with a graduate student(s) or the Principal Investigator later. In addition, the library offered separate meeting rooms by reservation to any patron. Participants also completed brief surveys, self-administered, or administered by an interviewer. Patrons received USD 20 in Metropolitan Transportation Authority Day Passes for their time in individual interviews and USD 10 for focus groups. Light refreshments (breakfast food) were also offered during the focus groups. Finally, the fourth group of unstably housed patrons (*n* = 5) held in June 2019 helped validate the results by giving their reactions to the research findings.

### 2.4. Recruitment of Library Staff Participants

The Supervising Librarian referred ten persons working in the library for voluntarily in-depth interviews with a researcher. We refer to “library staff” as anyone directly employed by the City/Library or by an external organization but stationed in an office at the library. For example, six of ten were section managers or librarians hired by the library and directly served patrons, e.g., Children’s and Teen sections; Sciences; Access Services; I-Can Disability Center; Library shop. Two others were homeless outreach workers from a nonprofit organization stationed in the library’s Homeless and Mental Health Outreach Office. Another volunteered for the Veterans Resource Center at the library. A fourth, employed by an outside nonprofit organization, worked in the library’s Workforce Development office. They were selected from 100 staff at the SDPL Central Library; half of the 100 staff worked directly with library patrons. Those interviewed also filled out the one-page socio-demographic survey. Four more library staff sent in their surveys when the library manager solicited their interest by email. However, they were not interviewed because their identity remained anonymous. Library staff received USD 10 Starbucks gift cards at the end of the in-depth interviews as token gifts for their time.

### 2.5. Research Measures and Instruments

The interview guide content for the interviews with the staff and patrons and the focus groups with patrons included open-ended questions regarding their experiences with housing instability, substance use, human trafficking, and their views on solutions such as having a social worker at the library. The library especially wanted to understand the barriers to meeting the aspirations of the patrons or what gets in the way of their educational needs. They also wanted to know (1) how many patrons overdosed at the library, (2) how the patrons obtain Narcan or have trouble accepting it, (3) why several overdose at the same time of the month (e.g., “five had a seizure on a Friday”), and (4) the types of drug availability/use and trends (e.g., fentanyl added to other drugs). Therefore, we incorporated these in the interview guide and survey.

### 2.6. Purpose and Content of the Brief Quantitative Survey

A total of 63 library patrons (*n* = 49) and library staff (*n* = 14) also completed brief, one-page surveys. The purpose was to gather socio-demographic data on the participants and triangulate the data from the in-depth interviews and focus groups. Surveys were self-administered in the interviewer’s presence during the in-depth interview or focus group.

The survey questions and response choices (the same for both patrons and staff) were created in consultation with the supervising librarian. Along with socio-demographic items, they included the following questions (response options included check all that apply and other):(1)What do you think are the educational needs of the library patrons who face housing/food insecurity, substance use, human trafficking, or other issues?(2)What further training do you think librarians need to help library patrons who face housing/food insecurity, etc.?(3)What changes in government or library policies are needed for library patrons facing homelessness, substance use, or human trafficking?(4)Would you like library patrons to participate in a peer leadership training (at the library) for those facing homelessness?

Socio-demographic write-in variables included age, gender identity, sexual identity, race/ethnicity, occupation, education level, housing status (sheltered, unsheltered, in transit and options for subsidized housing such as single room occupancy hotels and Section 8), and number of children/dependents. Disability was categorized as Yes-No with physical, mental, or educational options presented. U.S. Veteran status and U.S. citizenship were given Yes-No options.

### 2.7. Data Analysis and Dissemination

Research field staff trained on ethical study protocols audio-recorded interviews with informed consent from the participants. Audiotapes were transcribed verbatim without identifiers (e.g., names, names of agencies, or locations). Inter-coder reliability across coding was reached via a standard approach from Carey et al. [27]. The research team analyzed the data using a thematic analysis approach in which themes emerged.

In addition to eliciting feedback from the library patrons in the fourth focus group, we also presented the results to the Executive Team and Library Managers in August 2019, just days after a patron completed a suicide inside the library lobby, falling from the upper floor.

## 3. Results

### 3.1. Socio-Demographics

Table 1 displays the socio-demographic characteristics of the library patrons who participated in the research; 43% of the patrons were 50+ years old; the gender breakdown was 51% males and 49% females (none identified as gender-neutral or transgender); 47% were White, 25% Latino, 16% Black/African American. The majority were heterosexual and U.S. citizens; 22% lived in a shelter with the rest homeless or “in transit;” 29% had children; 33% identified as having a disability (physically or mentally); 16% were U.S. Veterans.

### 3.2. Views on the Role of the Library

In qualitative interviews with the library staff and patrons, themes emerged about the library’s role in housing insecure patrons’ lives. Participants commented on accessing the restrooms, study rooms, central air-conditioning, water fountains, books, computers with internet access, Wi-Fi, and a place to charge their phones. They also found it to be a safe and accepting place for resources.

*I can just isolate myself. Grab a book or even a newspaper and just tune everyone else out. You know? I’m safe and don’t have to worry about getting robbed.*—Male patron

*I’ve read a little bit in there and used the restrooms….I’ve also used the little outlets to charge my phone when I have a phone. I’ve also went to the 5th floor, and they got a little job center…the staff in there are outstanding…All you gotta do is spend a half-hour online…to make a resume…cover letter and they get information about you…it’s a good program. And they are really helpful. Too helpful. They don’t give you reasons or excuses.*—Male patron

Those living in shelters or on the street had few other places to go during the day. They often experienced stigma or exclusion from other public spaces. Barriers to government and community services included eligibility restrictions, long wait lists, and transportation costs.

*…and I know it makes the library--don’t look very appealing because there’s a lot of people hanging out outside. However, where can they go? The only place that lets their residents stay in the Tent. They can stay all day. That’s the only place they can spend the entire day there. However, the rest of the shelters… we can’t. And during the day, we’re all homeless, regardless if we have a place to sleep at night. During the day, we’re all homeless still.*—Male patron

The library strives not to discriminate. Patrons were only expelled if they actively used drugs on the premises or became assaultive. They might only be confronted if they were disruptive in other ways (e.g., snoring loudly). For example, library staff gently woke patrons to ensure they did not overdose on drugs.

*“You know, I think it’s managed quite well with the homeless issue and everything. Yeah, who would have heard, of everything, you go to the library. Yeah, well, back in my day, you’d find refuge at the library. Yeah, I found refuge in the library when I was in trouble at home.”*—Male patron

Many patrons were often unaware of services at the library and in the community. Some of the most vulnerable patrons facing multiple issues (homelessness, unemployment, financial, and food insecurity) found relief in discovering services they were unaware of. They often heard from a peer about a library feature.

*This library is awesome, honestly. It has all we need. I didn’t know until recently, but every time I come, I find new things that I didn’t know they have. Like this right here [meeting rooms]. And then last week, I found out about the workforce development. I wasn’t aware of it.*—Female patron

### 3.3. Recommendations for interventions

The following themes emerged as recommendations from library staff and patrons for interventions at the library: connect patrons to services; provide crisis prevention and intervention for mental health issues; create consistent support; and use peer advocates.

#### 3.3.1. Connect Patrons to Services and Each Other

*“There needs to be a professional who is just a social worker…who’s also part of the payroll for the library so that they’re an official library employee…rather than having to work through the bureaucracy of ‘…we’re just here in the office for the week, and we’re not really a staff member, and we have to follow a different protocol.’ ”*—a Library Manager

Most library staff participants (13 of 14) echoed the sentiment that having a social worker consistently at the library, even on weekends, would help with crises and follow-ups for those who need housing and other services beyond the role and training of a librarian. In the past, the library had a full-time peer (formerly unhoused) outreach worker who handled crises and helped patrons with case management referrals and follow-ups. He was hired by a nonprofit agency that supervised him. However, private funding for the position ran out after 2–3 years. At the time of this research, San Diego County funded three nonprofit organizational employees to work part-time at the library.

One librarian discussed the time a “social worker” connected with library patrons, freeing up the library staff’s time so they could focus on their job duties.

*“I mean, it’s such a large facility. You have that many people coming through here daily. Why isn’t this a focal point?”*—Library staff

Librarians often found themselves *“blindly reaching in the dark”* to assist with challenges that took away from their work at the library or were outside of their training. They shared resources with patrons but were then told those agencies denied services to the patrons for various ‘complicated’ reasons. These systematic barriers were outside the scope of the staff’s expertise and would ultimately leave a crisis or need unresolved. Instead, the staff expressed a need to *“refer to someone who really knew what they were doing.”*

Patrons also said they would attend group meetings at the library, e.g., for women or substance use recovery, such as a Narcotics Anonymous meeting, which a social worker could help facilitate. A patron talked about strength in connecting, showing potential to build on these connections:


*“Where everybody congregates and gives out bread someplace. Or everybody lets friends know where there’s gonna be a homeless dinner or something. So there’s definitely camaraderie.”*


The following are some connections a social worker could help the library expand on their links of patrons to services: housing linkages, substance use harm reduction and treatment, helping food insecure youth and families, and human trafficking/sexual exploitation services.

#### 3.3.2. Housing Linkages

Patrons also requested one-on-one assistance with computers and searching for apartment listings, which required allocated and dedicated time that the library staff did not have. Library staff emphasized a need for someone knowledgeable of resources and services; a library social worker’s “main focus of work” would be to *“deal with and recognize [what] the patrons would need.”*

*“It would be so wonderful to have someone here daily—that we could consistently refer people to. They could refer people out to all the available services. Because as you well know, folks don’t all of a sudden turnaround, and everything is better…”*—Library staff

For many patrons, barriers to services include travel, time, and convenience. One emphasized the need to make services *“more accessible where it’s one-stop.”* However, like many in his situation, he traveled through numerous hoops to access housing, but due to misinformation about eligibility, he stayed housed in a tent.


*“They told me here at the [City’s Tent shelter] that nobody will give you housing assistance without a verifiable income. So, I just gave up and stayed there [at the Tent]. And then, when I finally got there last month, she’s like, “Oh, yeah, we have programs for people like you. You don’t need any security deposit. Don’t worry about your bad credit. We pay your security deposit. And don’t worry about your rent for three months. Then, after three months, we’ll see how you are.” I’m like, “Why didn’t they tell me that then?.. if more people knew that, I guarantee you, and you push them, you’d get a lot more people out of that tent… when it rained a few times, I mean, it was miserable.”*


Stories such as his were familiar. He said having a social worker at the library would help. The library could be a convenient and strategic hub for information, referrals, and services.


*“I think that’s a great idea. One person who can do multiple things. Who’s a constant that you can go to. And once you see that it’s not –like you get over those hurdles, I think that’s a big thing. Because when I was at “the Tent”, I just thought, “I’m going to be here until I die.” I mean, not to be dramatic, but I just thought, “I’m never…going to get out of here.” …unless I win the lottery with some little money I have, yeah, you’re totally hopeless.”*


#### 3.3.3. Substance Use Harm Reduction and Treatment

In light of the acute opioid overdose crisis occurring within and surrounding the library, some patrons wanted to know how to access rehabilitation services and medically assisted treatment for their substance use. Library staff and patrons suggested the library host support groups, such as Alcoholics or Narcotics Anonymous, and education on harm reduction, such as safe drug use practices and safe injection practices, wound care, medically assisted treatment, and Naloxone training. Patrons had a varied level of awareness about identifying a drug when it contained a deadly dose of fentanyl (50–100 times more potent than morphine). Many patrons reported not knowing where to go for services or only knowing places that would not be helpful for their recovery. Many patrons also felt the library could be a place for general information and help for those struggling with substance use and overdoses and were open to the idea of working with a social worker. One male patron recounted:


*Yeah, that’s me. I just fell out a few weeks ago. I was drinking a little bit, then I did a big issue, and my friend didn’t know that it was so potent, and I just went out, yeah like I said, stopped breathing and everything…Yeah, we called 911 quick. He [friend] was trying to pound on my chest, but it was beyond his realm of medical training. So the paramedics rolled up, started beating on my chest, and got it going again. I woke up at UCSD [hospital].*


#### 3.3.4. Help Youth/Families

Although the McKinney-Vento Homeless Assistance Act helps youth facing homeless with their educational needs, barriers persist for many, causing library staff to intervene. *“Some kids have trouble with transportation to school. Or their parents start doing the paperwork, but do not finish it, so the children end up not going to school,”* said one staff member. A library staff member called a truancy officer once to talk to the youth.


*“And I want to say probably two or three days later, that particular kid came up to me, and I thought that kid hated me by how he was acting when the police officers were there. And he said, “Thank you. I really appreciate it. I’m in school now.”*


The library staff also noted “constant food insecurities” among youth: *”I cannot tell you how many times in the teen center we’ve been hearing the teens say, “I am hungry.”* Addressing food insecurity included providing information on adequate nutrition and especially helping teens who struggle in their relationship with food. For example, youth and teens’ mental wellness was affected by their struggles with stable housing. Signs of depression were overconsumption or underconsumption of food and oversleeping or insomnia. In an interview, one library patron admitted to “starving” while pregnant as a teen: “*I was starving for 3–4 months. I would not eat. I would constantly just sleep.”* She stated that this affected her son, but *“he was okay.”*

#### 3.3.5. Connect to Human trafficking and Sexual Exploitation Interventions

Several male and female patrons shared stories about witnessing sexual exploitation and suspected human trafficking around the library.

*I’ve seen sex traded to help get them into another camp. The girl is the penetrator, and then her whole crew follows. She’s the bait. She’s the one that goes in so they can all follow…It’s usually in exchange for their goods or whatever resources that they have. I know a lot of young ladies that have fell prey to that. The streets open up, and they swallow them*.—Male patron

Several female patrons described witnessing young women disappear, especially when a man living in a luxury high-rise building would go to the women’s shelter to offer a place to stay. The shelter and library later restricted him from coming. However, the patrons felt someone could do more to prevent these occurrences or help those victimized.

One male patron described constant offers to be an interstate trafficker of people or objects or to have sex in exchange for a place to stay. Sometimes he felt men were targeted for sexual exploitation even more than women.

The library created a human trafficking awareness program after a former Teen Center manager noticed a teen sleeping in the library study room one day. Sleeping used to be against the rules of conduct, *“so she asked about it, and it turned out someone was trafficking the teen and had put her there.”* The library was the only place and time the trafficked teen could sleep because she was working otherwise. Through the program, the staff were trained to be more aware of human trafficking signs and access information and resources to disseminate to the public. A social worker specialized in this area, especially a survivor of human trafficking, could help provide more targeted interventions for this population.

#### 3.3.6. Provide Crisis Prevention and Intervention

The SDPL Central Library has seen an alarming number of incoming patrons in crisis, with “daily incidents” occurring, leading to at least one patron getting suspended every day from the library. In addition, the staff observed an urgent unmet need regarding crisis prevention and how to intervene.

*“If someone has a mental health crisis, that conflicts with our rules of conduct”*—Library staff.

Some patrons with unstable housing said library staff could not respond to the needs of those facing homelessness and mental health issues, as their role and tasks are not centered on the specific needs of a crisis, mental health, substance abuse, or homelessness. *“A social worker in the library would be great, any kind of situation, to have someone who cares and can help.”* The perception of a library patron that library staffers are just there to say “*shhh*” is broken when staff listen and help them with their concerns. However, according to staff and patrons, it would also be helpful to have a dedicated clinical social worker on the library staff for librarians to refer patrons who could use longer-term or more in-depth professional mental health help and guidance, especially in times of crisis.

Often, crisis intervention turns to the police. However, the underlying tensions between law enforcement and those struggling with stable housing can prevent a patron in need from seeking the appropriate services, being protected, or managing an issue before it becomes a crisis.

*“You need to have people in here that can spot the signs even before something escalates…that a lot of them face a lot of traumas due to sex trafficking…or abuse. And when you try and tell the kids….’Hey, you need to report that’…they’re afraid of the police.”*—Library Manager

A young patron in the focus group said he would administer Narcan to others or himself without calling the police due to not wanting to get law enforcement involved. He believed that overdose cases were much higher than the reported cases due to others avoiding calling 911.

One library staff identified themselves as *“informational professionals. We’re not medics, social workers, life coaches.”* They will be notified of a crisis at the library *“when somebody downs a bottle of booze. Now, they’re screaming at somebody or nobody. Now, we got to get him out of here. Somebody starts a fight, shooting up in the restrooms, or whatever. We have a hard time being proactive.”*

#### 3.3.7. Create Consistent Support

Regarding rapport, library staff members emphasized the importance of consistency.


*“I think that more consistency is of value… I would like to see the County Health and Human Services in here…because we are at such an epicenter of what’s going on. We get a lot of the folks that can’t deal with the shelters, or the regimen of a shelter, or something like that. At one point, we got a grant…We had a gentleman that was here five days a week. He was able to gain rapport with a lot of folks. I remember a particular one, he finally got into a substance abuse program. He was able to find temporary housing. However, it was a lot of false starts to get to that point…but because he was consistent, non-judgmental, there to listen to him, he finally got him to that next stage.”*


In addition to training, staff members and patrons emphasized the necessity of having full-time social workers at the library seven days a week.


*“Now, we have three days a week [outreach workers at the library at the time]. However, I couldn’t tell you which organization is on which day – Without having to look that up. It’s Monday, Wednesday, and Friday, or excuse me, Monday, Tuesday, and Wednesday. But, okay, so what happens if somebody on Saturday needs some help? That was with that older lady. She’s 90 years old. We have a bunch of librarians calling 2-1-1, Father Joe’s, and calling anywhere they can to try to find some services for her. So that’s the changing nature of public service and our profession. So yeah. You got to be flexible.”*


Some patrons with unstable housing said library staff could not respond to the needs of those facing homelessness and mental health issues because their roles and tasks are not centered on the specific needs of such a crisis.

*“A social worker in the library would be great, any kind of situation to have someone who cares and who can help”*—Male patron

*“A lot of the residents at [the women’s shelter] come daily to the library. So, of course, that would be very helpful [having a social worker daily at the library].”*—Female patron

Not having stability and a consistent place to stay was typical.

*“Like if you’re in a shelter, you have six days or three days, then you’re off to another shelter.”*—Male patron

*“See, it’s the lack of stability that’s killing me. If I had something stable, I’d be fine.”*—Male patron

#### 3.3.8. Train Peer Leaders

Table 1 showed how 71% of the patrons answered yes to the question, “Would you like library patrons to participate in a peer leadership training (at the library) for those facing homelessness?” The rationale for asking this question was to see if unstably housed individuals had the potential to mobilize as a community as part of the strength/asset-seeking consensus organizing approach.

*“If there is an opportunity for their voice to be heard, I think many clients would take that because they are not very much given that platform. They are not granted autonomy in virtually any circumstances. So any situation where they do have that choice is huge.”*—Staff co-located at the library

*“A perfect example was the Hepatitis A outbreak in November 2017. It was a county-wide situation where everyone mobilized and wanted to be out there helping. Even the population [facing homelessness] themselves said we do need this. We do need a place to go to the bathroom at night and a place to wash our hands. Those leaders in that population say this is what we need. We don’t have housing, sure, but we still have these certain needs that aren’t being met.”*—Library staff

In addition, library patrons and staff appreciated people with lived experiences. They identified an individual who had struggled explicitly with homelessness in the past and served as a vocal board member of the Regional Task Force on the Homeless. He tries to “*bring light to some of the issues that maybe those board members don’t see*,” a patron recounts.

One person working at the library said the best youth advocates experienced and came out of homelessness. He said when placed in front of the right audience, “*suddenly everyone cares…Everyone wants to hear. However, if it’s somebody every day in the streets, they’re not going to listen to you. However, they are taken seriously because they’re like “this youth came to Washington—to tell you what’s wrong.”*

Service providers/library staff have also observed that patrons “*value the library so much they’re almost an extra set of eyes and ears*” for the library. If an issue arises, the patrons will help because they do not want the library to have problems. Library staff said, “…*it’s a collective thing. If something happens, it breaks up the whole idea of a safe environment… they want to preserve the peace*.” Many thought that having both a peer leader and a social worker would work best at libraries.

### 3.4. Survey Results: Perspectives on Training Needs for Librarians 

The survey asked, “What further training do you think librarians need to help library patrons who face housing/food insecurity, etc.?” Survey results showed that 63% of patrons and 86% of library managers said more training for librarians on detecting the signs of mental health issues would help (Table 2). For example, 48% of the patrons and 71% of the library employees agreed that there needed to be more education on substance use recovery and interventions. About 54% of patrons also thought it essential to train staff on what to do when they have substance use issues, and 57% of the library staff agreed.

However, some patrons said librarians should not be responsible for being educated on substance use interventions or issues because other professionals are already trained to handle these issues. More should not be added to a librarian’s job description. They thought it would be fairer to have a person trained in mental health and substance use at the library, such as a social worker, rather than asking a librarian to work outside their scope of practice.

Library staff reported that overdoses occurred monthly at the library, but it was just the tip of the iceberg. Focus group participants said overdoses happened as much as weekly or daily in the surrounding streets. Some patrons and staff attribute this to the increase in fentanyl in substances. Others felt that overdoses were high when people felt more hopeless. When asked if librarians should have more training on detecting and getting help for a patron needing life-saving measures, such as Narcan use, 54% of patrons agreed, and 50% of the library staff agreed.

Regarding training for library staff on substance use issues, an outreach worker at the library said:


*“They asked us about drugs, like what drugs look like and what someone high--what under the influence looks like, when someone has used meth versus cocaine versus marijuana. Most people know what someone intoxicated looks like… stuff that’s sort of second nature for us because we see it so much… is very unfamiliar and unsettling to many people, understandably so.”*



*“Same thing with serious mental illness… just knowing what someone in a psychotic episode experiencing mania-- paranoia looks like… and when to start getting concerned… it’s never a bad idea to have more training on suicidal ideations, plans, actions, and how to intervene.”*


Patrons indicated “other” training needs for librarians (they hand-wrote these in their surveys). Examples are “more knowledge of resources, socialization, social contact, self-esteem, sensitivity training, just being able to relate, be human.” Others thought librarians could be trained on homelessness, autism, addressing children and adult obesity, seeking safety, depression without a substance use disorder, and training on how to spot crises such as suicidality or overdoses. They also recommended having medical staff on-premise.

Staff “other responses” included: finding housing and employment; having trained personnel present in the library such as social workers, drug and alcohol abuse counselors, and law enforcement officers; information for services to refer hungry and homeless; learning about the diseases persons “homeless” are prone to; safety for library staff; and workforce programs.

Staff at the library had mixed perspectives about whether librarians should be expected to deal with these issues. Some said “That is not the function, job, or purpose of librarians” or “We are librarians, not social workers or nurses.” Others said, “Librarians should have the information to refer these patrons to appropriate organizations and services.” 

### 3.5. Survey Results: Perspectives on Educational Needs for Patrons

For the perceived educational needs of library patrons (Table 3), patrons (67%) and library staff (64%) concurred with the need for patrons to access electronic devices and access information about financial assistance for education (63% patrons; 57% library staff). However, patrons (47%) agreed less about the patrons’ need for substance use recovery/interventions than what the library staff thought (71%), and similarly regarding human trafficking (31% patrons, 57% library staff). A lower percentage, 24% of patrons and 29% of library staff thought patrons needed education about HIV or other health issues.

### 3.6. Survey Results: Perceived Changes Needed in Government or Library Policies

The greatest perceived policy change need was a permanent social worker in the library (69% of patrons, 93% of library staff), followed by information about financial assistance for education and substance use recovery/interventions (Table 4). For example, one staff suggested an onsite social worker but said, *“the City needs to step up, not the library.”* Instead, a patron said*, “give us programs and options and social workers.”* Library staff identified the need for changes in patron employment, and a patron wrote, “MTS’ (transportation authority) better identifying the mentally disabled.”

Library staff noted “other” the need for “sensitivity training for library staff.” Library patrons mentioned “other policies“, “raising the minimum standard of conduct/enforcement”, “confidentiality”, and “treat us with more respect and kindness.”

Library staff noted the need for “free housing,” and a patron wrote “housing wait lists.” Patrons mentioned changing the policies on “being treated as criminals and having no place for people to live” and suggested having “more information on shelter and housing availability, education, and where to go.”

Library staff also wrote for “other” that police treated persons facing homelessness unfairly compared to those who were not homeless. In addition, patrons noted “other” policy change needs: “police not taking property and always moving them from place to place” and “private security having zero tolerance.”

## 4. Discussion

Findings from the interviews and surveys with the library patrons and staff in this study demonstrate a unique opportunity to conduct assessments and interventions in a space where many persons struggling with housing instability and homelessness congregate. The study shows that library patrons with homelessness, mental disabilities, or substance use issues willingly met for in-depth interviews and focus groups at the library in light of the barriers they experienced, such as housing and transportation. Most (69% of patrons; 93% of staff) felt having a permanent social worker at the library would help to facilitate unstably housed patrons’ connections to resources. In addition, the top librarian training need was “assisting patrons with mental health issues.” Still, most agreed that libraries need more than staff training to de-escalate a mental health crisis.

For example, the library patrons and staff agreed that training librarians on responding to patron mental health crises could help. However, this requires de-escalation and counseling techniques, which clinical social workers are trained in. Libraries can leverage training on detecting and managing signs of a mental health crisis as a bridge for patrons afraid of law enforcement or who do not meet the eligibility criteria for services at other agencies or shelters. They can then refer the patron to a social worker on staff who can work sensitively with the library security. Training also allows librarians and staff members to participate in preventive services proactively before a crisis escalates. Other studies have found that a patient-focused case-management approach, such as those offered by social workers, is beneficial for long-term treatment for this vulnerable population [18].

Library patrons who face mental health crises, violence, human trafficking, and sexual assault on the street need more targeted interventions [28]. For example, studies show that those who are unstably housed and use opioids, fentanyl, and methamphetamines, benefit from “housing first” models for recovery [29,30]. Library patrons who used substances emphasized needing someone at the library who understands and relates to their needs and circumstances [29]. The best scenario is to have multiple social workers or mental health staff and interns on-premise to ensure someone is present as situations arise at any given time.

Other public sectors, e.g., police in Indiana and Minnesota, have increasingly engaged social workers to help them tackle the grand challenges of homelessness, mental illness, and human trafficking [31,32]. In addition, peer health navigators have enabled populations in other settings to navigate medical systems [33]. Social workers, peer navigators, and outreach workers could work together on site at libraries and other public spaces.

Not everyone in this study found it necessary to have a social worker at the library nor to have librarians trained on administering Narcan for opioid overdoses, for example. The majority of public service staff are represented by a labor union. They have yet to agree to permit designated staff to carry Narcan. However, with escalating overdoses at the SDPL Central Library, the Library Department’s Executive Team, consisting of seven unrepresented/unclassified City employees, has had training and carry Narcan. The lead contracted Security guards also carry Narcan and have saved multiple lives. They are in the process of negotiating the new security contract and hoping to have all guards carry Narcan and have training to properly administer it. Many library staff have expressed the desire to voluntarily receive training and carry Narcan should the need arise and have advocated to their union representatives.

Telemedicine is also a potential intervention at the library [34,35] for patrons who have trouble finding transportation to a medical or mental health session. In addition, some libraries, such as the SDPL Central Library, have private study rooms and laptops all patrons can reserve. Access to the internet was one of the most pressing needs identified by library patrons in this study and others, e.g., in Michigan [2], and could be leveraged with interventions.

The library patrons and staff in this study concurred with each other about the persistent educational needs of patrons, consistent with other studies [9]. For example, participants identified the need to connect unstably housed library patrons with educational resources such as General Education Development (GED) tests or high school diploma equivalent and workforce development resources. In addition, the SDPL Central Library reduces the digital divide by providing access to technology and computers. However, further workshops or individual assistance could be offered to patrons navigating employment and social service applications, which patrons found challenging to complete independently. For example, a social worker or case management team, including college student interns, could help with these tasks.

The limitations of this study include its relatively small number of participants (*n* = 63), despite the number being large for a qualitative study. It was confined to the study of one library in one city. However, the study occurred at the largest metropolitan library in San Diego City and County and may mirror what many large urban libraries encounter across the U.S. For example, the recruitment of participants relied on those who responded independently to fliers, were approached randomly on the street, or received a referral from someone else. We did not purposively outreach to less visible or more challenging to reach populations in the farther perimeters of the library or other parts of the city. For example, we did not get a representative sample of those with different gender identities or youth who may have other perspectives to add. The validity and reliability of the data could be tested in a more extensive survey; the brief one-page questionnaire was intended to augment the participants’ qualitative interview responses and provide socio-demographic data, not to conduct quantitative survey data analysis. However, many of our findings were similar to other perspectives in the literature on the societal barriers unstably housed library patrons encounter [9,21].

## 5. Conclusions

Findings may inform policy and practice at other libraries nationwide experiencing similar issues, namely opioid overdoses, housing instability, and mental health issues among patron populations. Daily, library patrons present with crises both within the library walls and on the outside perimeters. Counseling programs, e.g., one associated with retention in a medication-assisted treatment program for people with opioid use disorder [36], could be offered in public spaces such as libraries via telehealth, but needs to be tested out. In addition, staff trained in crisis intervention, such as social workers, could collaborate with other disciplines (e.g., nursing and public health) to make a collective impact on site [37]. Other studies have called for the expanded role of public health and social work at public libraries across the nation [3,4,5,6,7,8,9,10,11,38].

The library presents an opportunity to find and serve some of the most hidden, vulnerable populations in the U.S. who often fall off the radar, think they do not qualify for services, or do not know where to get help. For this reason, libraries may offer a space unlike few others for those thriving or struggling to survive.

## Figures and Tables

**Table 1 ijerph-19-08449-t001:** Socio-demographic characteristics of library patron survey participants (*n* = 49).

Category	Response	N (%)
Age (20–70)	18–25	5 (10%)
	26–34	9 (18%)
	35–49	14 (29%)
	50–64	19 (39%)
	65 or Older	2 (4%)
Gender Identity	Male	25 (51%)
	Female	24 (49%)
Race/Ethnicity	White	23 (47%)
	Latino/x	12 (25%)
	Black/African American	8 (16%)
	Other	6 (12%)
Sexual Identity	Heterosexual	45 (92%)
	Lesbian/Gay/Bisexual	4 (8%)
Housing	“Homeless” (or “Transit”)	31 (63%)
	Living in a Shelter	11 (22%)
	Couch surfing or Staying with a friend	3 (6%)
	SRO/Section 8 or senior studio apartment	3 (6%)
	Soon to vacate	1 (2%)
Children/Dependents	Yes	14 (29%)
	No	35 (71%)
Disability	Yes	16 (33%)
	No	33 (67%)
U.S. Veteran Status	Yes	8 (16%)
	No	41 (84%)
Citizenship	U.S. Citizen	47 (96%)
	Non-citizen	2 (4%)
Would you like library patrons to participate in a peer leadership training (at the library) for those facing homelessness?	Yes	35 (71%)
	Maybe	1 (2%)
	No or blank	12 (24%)

**Table 2 ijerph-19-08449-t002:** Perceived training needs of the librarians (*n* = 63; 49 Patrons, 14 Library Staff).

Training Needs of Librarians	Patrons	Library Staff
How to help patrons experiencing mental health issues (e.g., suicidal tendencies)	63%	86%
How to detect and get help for a patron needing life-saving measures (e.g., Narcan)	53%	50%
How to help patrons with substance use issues	51%	57%
How to identify, detect, and intervene with human trafficking	51%	50%
Training about HIV or other health conditions, e.g., Hep A	45%	29%
Other	12%	57%

**Table 3 ijerph-19-08449-t003:** Perceived Educational Needs of Library Patrons (*n* = 63; 49 Patrons, 14 Library Staff).

Educational Needs of Patrons	Patrons	Library Staff
Access to electronic devices	67%	64%
Access to information about financial assistance for education	63%	57%
Substance use recovery, interventions	47%	71%
Human Trafficking	31%	57%
HIV or other health issues	24%	29%
Other: (specify)	47%	79%

**Table 4 ijerph-19-08449-t004:** Perceived Changes Needed in Government or Library Policies (*n* = 63; 49 Patrons, 14 Library Staff).

Changes Needed in Government or Library Policies	Patrons	Library Staff
Having a permanent social worker in the library	69%	93%
Shelter and housing availability	67%	79%
Treatment by police or others	59%	36%
Substance use treatment options	53%	57%
Policies for formerly incarcerated	29%	29%
Other	24%	21%

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
