# Peer review of "Homelessness at the San Diego Central Library: Assessing the Potential Role of Social Workers"

_ijerph, 2022, doi:10.3390/ijerph19148449_

Round 1
Reviewer 1 Report
This is an important and timely topic of study and library/social work collaborations are an emerging area of practice in both the social work and library worlds. I appreciate the authors' work in this area and their attempt to address a local need. I especially appreciate them creating this as a class exercise to give students relevant practice with connecting research to the "real world." However, this manuscript needs quite a bit of work to make it appropriate for publication. This still reads as a class assignment/paper, focused specifically on San Diego Public Library, rather than a publishable research manuscript which should 1) better integrate existing research with this project and explain how this project flows/results from the previous work (there's a growing body of work about library patron and staff's needs re: homelessness, mental health, substance use, etc.) , 2) build upon the current body of work in the area of library social work or libraries and homelessness, etc., filling a gap in the literature and expanding our knowledge on the topics at hand 3) make a clear case for why this project is relevant to other libraries around the country (which is essential for publication). There seemed to be a significant disconnect between this project and the existing research on the topics covered in this project, other than some brief statements at the beginning and the end that quickly refer to existing literature.
Also, the methods are unclear in places with inconsistencies present. It is unclear whether this project was approved by an ethics board/IRB. Recruitment methods are unclear in places (for example, how did the research team determine who was "addicted to opioids" and eligible to participate in one of the groups?).
The title and abstract are a bit misleading. This seems to be a project focused specifically on how social work can help with homelessness, etc. rather than a general assessment that looked at a variety of options for helping the library. This should be clearer in the beginning since it seems to be assumed that a social worker should be in the library and then questions were focused on how they might be able to help.
Overall, I support the authors in their efforts with this project but think the manuscript needs a significant amount of revising to make it appropriate for publication.
Author Response
Please see the attachment.
Responses to Reviewer 1
Reviewer Comment:
- better integrate existing research with this project and explain how this project flows/results from the previous work (there's a growing body of work about library patron and staff's needs re: homelessness, mental health, substance use, etc.) ,
- build upon the current body of work in the area of library social work or libraries and homelessness, etc., filling a gap in the literature and expanding our knowledge on the topics at hand
- make a clear case for why this project is relevant to other libraries around the country (which is essential for publication).
Author response: Thank you! We did a major overhaul of the paper both in the Introduction and Discussion to better ground the paper in the context of other studies.
Reviewer Comment: It is unclear whether this project was approved by an ethics board/IRB.
Author Response: Thank you- we added that the study was approved by the university’s institutional ethics review board.
Reviewer Comment: Recruitment methods are unclear in places (for example, how did the research team determine who was "addicted to opioids" and eligible to participate in one of the groups?
Author Response: Thank you- we elaborated on our recruitment methods more. For example, we described how graduate students (doctoral students with lived experiences with substance use) were able to detect and approach participants on the streets to get them to join the focus group for those using opioids, for example.
Reviewer Comment: The title and abstract are a bit misleading. This seems to be a project focused specifically on how social work can help with homelessness, etc. rather than a general assessment that looked at a variety of options for helping the library. This should be clearer in the beginning since it seems to be assumed that a social worker should be in the library and then questions were focused on how they might be able to help.
Author Response: Thank you for all of the helpful feedback! We changed the title to: “Homelessness at the San Diego Central Public Library: Assessing the potential role of social workers.”

Reviewer 2 Report
This paper interviews patrons of a public library and library staff about service needs of patrons experiencing homelessness. The topic is important and the implications of the research are generally sound. However, some clarification in presentation of the results and methods are needed, the paper could use editing and the analysis is somewhat superficial.
In the introduction the authors claim that "Additionally, the library offers and safe and label free environment where the stigma falls off their shoulders as they walk in." No evidence for this statement is provided by referencing research. This contention is repeated in the discussion, "Patrons found the public library to be a safe haven..." but they provide no evidence for this in their interviews. These statements may very well be true, but they require some evidence to support them. Although they are free to access the library, they may feel stigmatized by library staff or patrons not experiencing homelessness.
In the methods, it is not clear how many individual interviews were completed with patrons and whether all were the same as those who participated in focus groups. Were focus group participants given an incentive as well? The recruitment and sample for the "library staff" is also confusing. The authors write "External potential assets in the community included service providers/organizations and organizations who provide services within the library." So, these were people who worked for other organizations but provided services within the library? How many participants in this category were recruited and out of how many potential participants? How were they identified and recruited? The sentence quoted above also has grammatical problems, "organizations who" should be organizations that.
There are also some grammatical problems with the description of the content of the interviews. Some items in the list are written as topics and others as questions.
The section "hiring a full time social worker" could use some expansion. What did library staff see as the main drawback to having only part-time outreach workers employed by outside agencies? Somewhere in the beginning of the results, it would be useful to provide some data about how participants view the library, i.e., why they use it, whether they feel stigmatized, judged or not, etc. It also seems that there is at least some ambivalence from librarians about their roles in serving the needs of people experiencing homelessness, with some saying that they are librarians, not social workers. In addition, not all agreed about the kinds of services, trainings and staff that the library should provide. For example, only 50% agreed that they needed training on life saving measures.
In at least a couple of places in the manuscript, the authors write about needing to provide "needle awareness." It is not clear what the authors mean by this. Harm reduction? Safe injection practices and wound care? Syringe exchange?
The quote about food insecurity needs more of an introduction or transition. I would also recommend re-organizing some of your paragraphs. In some places two different themes are in one paragraph and the continuation of the second theme follows in a separate paragraph (as, for example, in the discussion of food insecurity).
The sentence, "He attributed not calling 911 to the lower known cases of overdose versus the actual incidences." The subject and object seem to be reversed here. I think what you mean to say is that the participant believed that actual cases of overdose were much higher than the reported cases due to the fact that many opioid users will not call 911. Also, I think "incidents" is the correct word here.
Much of the results are presented as lists, presumably of open ended answers to the surveys. However, some of these items are very hard to understand. For example, the need for training on "autism for children with obesity" or "training on how to spot these." It is not clear exactly what "these" refers to. Other examples include "person to person specific", "step by step programs," something like the Alpha project which houses people under a large tent." Similarly, in a list of changes that patrons thought needed to be made, I assume that "being treated as criminals" referred to patrons wanting not to be treated as criminals, as it was phrased in other places in the list, e.g., "police not taking property."
In the discussion, the sentence "They can specifically assist those with opioid and other substance use addictions to get help for detoxification or rehabilitation or even harm reduction." Detoxification is not an appropriate response to opioid use disorder and, in fact, is contraindicated because of the high risk of overdose. Also the order of the list, and the addition of "even" before harm reduction makes it seem as if this is the least desirable choice. I would argue that it is the most desirable if the participant is not ready to stop using drugs. You might want to re-consider terms such as addiction. Substance use disorder is the more accepted term now.
Author Response
Please see the attachment.
Responses to Reviewer 2
Reviewer Comment: Some clarification in presentation of the results and methods are needed, the paper could use editing and the analysis is somewhat superficial.
Author Response: Thank you! We did a major re-write of the paper to address all of these issues.
Reviewer Comment: In the introduction the authors claim that "Additionally, the library offers a safe and label free environment where the stigma falls off their shoulders as they walk in." No evidence for this statement is provided by referencing research. This contention is repeated in the discussion, "Patrons found the public library to be a safe haven..." but they provide no evidence for this in their interviews. These statements may very well be true, but they require some evidence to support them. Although they are free to access the library, they may feel stigmatized by library staff or patrons not experiencing homelessness.
Author Response: Thank you, we’ve removed that language. We also added a section at the beginning of the Results with quotes to support the perspectives of the patrons about the library.
Reviewer Comment: In the methods, it is not clear how many individual interviews were completed with patrons and whether all were the same as those who participated in focus groups. Were focus group participants given an incentive as well?
Author Response: Thank you- we added clarification about the unduplicated number of participants in the Methods.
Reviewer Comment: The recruitment and sample for the "library staff" is also confusing. The authors write "External potential assets in the community included service providers/organizations and organizations who provide services within the library." So, these were people who worked for other organizations but provided services within the library? How many participants in this category were recruited and out of how many potential participants? How were they identified and recruited? The sentence quoted above also has grammatical problems, "organizations who" should be organizations that.
Author Response: Thank you! We removed that sentence from the manuscript and fixed the grammar. Under Methods p.4, we added details to clarify the staff/stakeholders interviewed:
The Supervising Librarian referred ten library staff/stakeholders for voluntary participation in an in-depth interview with the researcher/student research assistants. All staff/stakeholders were those working directly with library patrons and stationed in the Central library. Library staff were managers or staff in different library sections. For example, they were from the Children’s or Teen sections; Sciences; Access Services; I-Can Disability Center; Library shop. Others were non-profit organizational homeless outreach workers in the Homeless and Mental Health Outreach Office or working for the Veterans Resource Center, both located on the same floor of the library.
Reviewer Comment: There are also some grammatical problems with the description of the content of the interviews. Some items in the list are written as topics and others as questions.
Author Response: Thank you! We revised that entire section (p.4).
Reviewer Comment: The section "hiring a full time social worker" could use some expansion. What did library staff see as the main drawback to having only part-time outreach workers employed by outside agencies?
Reviewer Comment: Somewhere in the beginning of the results, it would be useful to provide some data about how participants view the library, i.e., why they use it, whether they feel stigmatized, judged or not, etc. It also seems that there is at least some ambivalence from librarians about their roles in serving the needs of people experiencing homelessness, with some saying that they are librarians, not social workers. In addition, not all agreed about the kinds of services, trainings and staff that the library should provide. For example, only 50% agreed that they needed training on life saving measures.
Author Response: Thank you! We added a section on the “views on the role of the library” to the beginning of the results section, p.5-6. Very few ever said they felt judged or stigmatized at the library, so we did not include any of those comments. In the Discussion, we added p.14:
Finally, the current study found a majority of staff and patrons believed having a consistent social worker at the library that patrons could follow up with at any time would be ideal. Not everyone found it necessary to have a social worker at the library or to have staff further trained on some topics (e.g. staff administering Narcan). However, the San Diego Central Library has developed a harm reduction approach for patrons suffering from opioid overdoses. They work with security staff to make Narcan available and since the end of the data collection for this study, some underwent training to carry and administer Narcan for patrons who overdose.
Reviewer Comment: In at least a couple of places in the manuscript, the authors write about needing to provide "needle awareness." It is not clear what the authors mean by this. Harm reduction? Safe injection practices and wound care? Syringe exchange?
Author Response: After revisiting the interviews, this was changed to “along with education on harm reduction, such as safe drug use practices like safe injection practices, wound care, medically assisted treatment, and Naloxone training.”
Reviewer Comment: The quote about food insecurity needs more of an introduction or transition. I would also recommend re-organizing some of your paragraphs. In some places two different themes are in one paragraph and the continuation of the second theme follows in a separate paragraph (as, for example, in the discussion of food insecurity).
Author Response: Thank you! We revised the section on food insecurity, p.8-9, and changed the title to helping youth/families.
Reviewer Comment: The sentence, "He attributed not calling 911 to the lower known cases of overdose versus the actual incidences." The subject and object seem to be reversed here. I think what you mean to say is that the participant believed that actual cases of overdose were much higher than the reported cases due to the fact that many opioid users will not call 911. Also, I think "incidents" is the correct word here.
Author Response: Thank you for pointing this out, this was changed to “A young patron in the focus group said he would administer Narcan to others or to himself without calling the police due to not wanting to get law enforcement involved. He believed that overdose cases were much higher than the reported cases due to others avoiding calling 911 for this reason. Library patrons disclosed the mental health impact of having lived on the streets for years, whether that’s struggling with depression or having a safe haven to call home.”
Reviewer Comment: Much of the results are presented as lists, presumably of open-ended answers to the surveys. However, some of these items are very hard to understand. For example, the need for training on "autism for children with obesity" or "training on how to spot these." It is not clear exactly what "these" refers to. Other examples include "person to person specific", "step by step programs," something like the Alpha project which houses people under a large tent." Similarly, in a list of changes that patrons thought needed to be made, I assume that "being treated as criminals" referred to patrons wanting not to be treated as criminals, as it was phrased in other places in the list, e.g., "police not taking property."
Author Response: Thank you- We edited the lists of “other” responses, p.13-14.
Reviewer Comment: In the discussion, the sentence "They can specifically assist those with opioid and other substance use addictions to get help for detoxification or rehabilitation or even harm reduction." Detoxification is not an appropriate response to opioid use disorder and, in fact, is contraindicated because of the high risk of overdose. Also the order of the list, and the addition of "even" before harm reduction makes it seem as if this is the least desirable choice. I would argue that it is the most desirable if the participant is not ready to stop using drugs. You might want to re-consider terms such as addiction. Substance use disorder is the more accepted term now.
Author Response: Thank you for pointing out the sentence structure and how it can make it seem that we are saying harm reduction is the least desirable choice when that was not intended. The sentence was changed to “They can specifically assist those with opioid and other substance use disorders in getting connected to services they are looking for, such as harm reduction tools, detoxification, and rehabilitation programs.” We hope that this sentence is more objective and shows the variety of options that patrons were looking for.
Any reference to addiction was also changed to substance use disorder.

Reviewer 3 Report
- This study used a sound qualitative study to examine the homeless issue in public spaces, however, the study title seem too broad to focus one specific needs. The author should re-examine this title issue.
- The study settings needed to describe more detail in the content.
- The interview consistency of the interview should describe in detail, and the validity and reliability of the data as well.
- The purpose and content of the survey is lacking to describe in the method and is it quantitative design? and the detail of the sampling should describe in detail.
- The analyses of qualitative data are appropriate and well described.
- the study limitations should add in the discussion.
Author Response
Please see the attachment.
Responses to Reviewer 3
Reviewer Comment: This study used a sound qualitative study to examine the homeless issue in public spaces, however, the study title seem too broad to focus one specific needs. The author should re-examine this title issue.
Author Response: Thank you for all of the helpful feedback! We changed the title to: “Homelessness at the San Diego Central Public Library: Assessing the potential role of social workers.”
Reviewer Comment: The study settings needed to describe more detail in the content.
Author Response: Thank you! We added a section on study setting to the beginning of the Methods section.
Reviewer Comment: The interview consistency of the interview should describe in detail, and the validity and reliability of the data as well.
Author Response: Thank you! We did a major overhaul of the paper and reflected on the validity and reliability limitations of the data under the limitations in the discussion. However, please let us know if we need to describe these methods more as it is an important point, but we are not sure if we addressed them. Perhaps you can give more of an example—consistency of the interview? We are also attaching both the interview guide and the survey which can be published with the article if it seems appropriate.
Reviewer Comment: The purpose and content of the survey is lacking to describe in the method and is it quantitative design? and the detail of the sampling should describe in detail.
Author Response: Thank you! We added the purpose of the survey, p.4:
The purpose was to gather socio-demographic data on the participants and to triangulate the data from the in-depth interviews and focus groups. Surveys were self-administered in the presence of the interviewer during the time of their in-depth interview or focus group.
Reviewer Comment: the study limitations should add in the discussion.
Author Response: Thank you- we added a limitations paragraph to the Discussion.

Round 2
Reviewer 1 Report
Thank you for working to improve the manuscript and to better relate it to the broader body of research on libraries, homelessness, and the role of social workers. The changes have made the manuscript much stronger. The topic is timely and of much interest in the library and social work worlds.
Author Response
Dear Reviewer 1,
Thank you very much for your positive feedback! We have revised the manuscript again and believe that the grammar/writing/spelling is improved in this version.
Reviewer 2 Report
While the authors have made some revisions to the paper, some problems were not resolved and new issues have emerged. For example, there appears to be a new section on "peer advocacy," but readers are offered no context to understand this section. What questions were asked to elicit these quotes? How was peer advocacy described to participants? Further, the organization of this section is confusing. The paper starts by describing the number who agreed they would be interested in attending a training for peer advocacy, then to discuss the high percentage who agreed to participate in future interviews as further proof of participants' willingness to undergo such a training. Agreeing to participate in the study does not reveal much of anything about their feelings about being an advocate.
Much of the write-in answers are still provided as a list, even when some of the things mentioned are not trainings but other things they wish could be changed, like changes in policy. In some ways, participants (both patrons and library staff) seem to be saying, by filling in policy-related answers, that training librarians is unlikely to make much of a difference in filling the needs of people experiencing homelessness. An explanation for the lower percentage of patrons agreeing that librarians needed trainings is now provided, but is poorly written and difficult to follow.
Many of the quotes in the paper go beyond what is needed to make the point the authors are trying to make, e.g. libraries offer safety, or having a social worker would help patrons navigate confusing eligibility criteria.
Some of the methods is still confusing. For example, the authors often refer to participants as either people experiencing homelessness (patrons) or librarians. However, it is clear from recruitment and some of the results that people who offered services but were not employed by the library were also included and only occasionally mentioned.
